# Multi-layer networks reveal changes in plant-bird interactions driven by invasive species

Jaume Izquierdo-Palma [1], Dailos Hernández-Brito [1] ✉, Fernando Hiraldo[1], David García-Callejas [2], José L. Tella[1] & Martina Carrete [3]

Biological invasions can strongly disrupt ecosystems, reshaping their structure and functioning. We investigate how two widespread invasive parrots –the rose-ringed parakeet *Psittacula krameri* and the monk parakeet *Myiopsitta monachus*- affect plant-bird interaction networks using a multilayer framework. Field data were collected over a full annual cycle in an area with both species, accumulating 288 h of observations and tracking 24,561 fruits from 576 plants. Parakeets modified networks by introducing novel interactions, increasing species turnover and altering modularity and nestedness. Acting as both seed predators and dispersers, they became central connectors, enabling native birds to access previously unavailable resources and increasing rare dispersal mechanisms. Their activities increased antagonisms and generated new interspecific interactions with numerous plant species. By exploiting plants not previously used by local birds, parakeets heightened the risk of secondary invasions and the spread of exotic plants. These findings underscore their dual roles in disrupting and restructuring ecological networks and stress the need to reassess their contributions in native and invaded ecosystems. Understanding their potential to facilitate exotic plant expansion is critical, as their ecological impacts will likely intensify with population growth and geographic spread. Comprehensive assessments are essential to predict and mitigate these far-reaching consequences.

Biological invasions are an increasingly common phenomenon in our interconnected world, showing no signs of saturation[1,2]. While some species naturally disperse beyond their native ranges[3], most current invasive species are almost exclusively introduced through human activities, crossing biogeographic barriers that would otherwise be insurmountable[4]. Thus, a hallmark of invasive species in the Anthropocene is their lack of evolutionary history with native species, often leading to disruptions in biotic interactions and ecological processes[5]. Invasive species often modify interactions among native species, altering community functionality, triggering cascading effects and compromising biodiversity and ecosystem services[6]. Invasive species can occasionally provide food or habitat for native species or restore essential services like pollination or seed dispersal (e.g., refs. 7–10). However, preferences of native and exotic species may differ and, for example, invasive animals may disperse exotic plants more effectively than native ones, favoring secondary invasions[11]. Given the complexity and variety of interactions that species can establish among themselves, even seemingly beneficial effects of invasive species may lead to negative outcomes for specific

components of the ecosystem[12] or produce subtle impacts that may go unnoticed but can promote unexpected long term changes[13].

Ecological network analysis provides a robust framework for understanding how species interactions shape community structure and functionality[14–16]. In general, mutualistic networks exhibit asymmetrical dependencies and nested patterns (i.e., specialists interact with species that form perfect subsets of the species with which generalists interact), whereas antagonistic networks are more modular (i.e., stronger connections between species within than between modules)[17,18] (but see ref. 19). However, classifying species as purely mutualistic or antagonistic can be both conceptually challenging and overly simplistic, as it may overlook the complexity of plant-animal interactions. For example, seed eaters may act as predators when destroying seeds (antagonism), but can also as dispersers by plucking whole fruits from the mother plant and dropping intact seeds at a distance after consuming the pulp or even some seeds (mutualism)[20]. Such context-dependent nature of species interactions suggest that mutualisms and antagonisms exist on a continuum rather than as isolated and distinct processes[21–23]. Furthermore, under certain circumstances, species may also

[1]Department of Conservation Biology, Estación Biológica de Doñana (CSIC), Seville, Spain. [2]Institute of Biology, University of Graz, Graz, Austria. [3]Department of Physical, Chemical and Natural Systems, Universidad Pablo de Olavide, Seville, Spain. ✉e-mail: dailoshb@ebd.csic.es

https://doi.org/10.1038/s42003-025-09130-4                                                    **Article**

facilitate resource use by others, creating novel interactions between previously unrelated components of a community[24]. Therefore, to improve our understanding and our ability to predict community and ecosystem responses to changes in species composition –whether due to biological invasions or biodiversity loss- it is crucial to consider systems in all their complexity[25]. The development of multilayer networks, which encode different interaction types in a single mathematical object, has enhanced our ability to study systems where species simultaneously engage in antagonistic and mutualistic roles, showing that network properties and resilience can change when different interaction types are considered together[26,27].

Birds play crucial roles in ecosystems as seed dispersers or pollinators (mutualisms) as well as predators of flowers and seeds (antagonisms)[28], being involved in tangled webs of interactions that provide important ecosystem services[29]. Birds are also key contributors to global change, with many species introduced to new regions via the pet trade[30–32]. Parrots (common name for birds belonging to the order Psittaciformes) are one of the most popular cage birds[31,33] and millions of individuals have been moved around the world for the pet trade for decades[34]. Accidental or deliberate releases have resulted in the introduction of at least 166 parrot species into 120 countries, with 60 species successfully establishing populations in urban and peri-urban environments worldwide[35]. Parrots possess unique morphological and physiological traits that allow them to exploit diverse plant resources, including a strong beak with the ability to move both jaws[36], zygodactyl feet and a fleshy and mobile tongue that allows them to access, manipulate and taste food[37], and a high physiological resistance to most phytotoxins[38]. Although parrots have largely been considered seed predators, these traits enable them to act as generalist herbivores and mutualists, participating in seed dispersal and pollination[10,22,39–42]. Parrots thus play important ecological roles in their native ecosystems, modulating the functionality of ecological networks[26,39]. This multifaceted role is also expected in areas where these species are invasive. However, while invasive parrot populations have been reported to cause economic impacts on crops mainly through fruit and seed predation[43], there is no information on their role as seed dispersers nor on the changes they may cause in the recipient ecosystems or the synergetic interactions they may establish with non-native plants.

We evaluated the impacts of invasive parrot species on the structure and functioning of recipient plant-bird communities using a multilayer network framework. To do this, we collected field data over a full annual cycle in an area occupied by the two world's most widespread invasive parrots: the rose-ringed parakeet (*Psittacula krameri*) and the monk parakeet (*Myiopsitta monachus*)[35] (Supplementary Fig. 1). Parrots are a diverse group of birds characterized by strong curved beaks, zygodactyl feet and vivid plumage. The term parakeet does not denote a separate taxonomic group but rather refers to small to medium-sized parrots with slender bodies and long, tapering tails. Rose-ringed and monk parakeets possess unique traits typical of parrots but distinct from those of birds in the recipient community, including their habit of discarding uneaten food that may provide native birds with access to fruits and seeds that would otherwise remain inaccessible[44,45]. Our work address two key questions and associated predictions. First, what role do invasive parakeets play in the multilayer network and its mutualistic and antagonistic subnetworks? We predict that the parrots' ability to predate on and disperse a wide variety of fruits and seeds will position them as central nodes within the network. This centrality and duality as mutualists and antagonists is expected to increase the network's nestedness and strengthen interconnections between subnetworks. Additionally, parrots are likely to alter the network structure not only by rewiring existing interactions but also by introducing novel interactions with plant species not previously utilized by the native species in the host community. Second, how do invasive parakeets interact with native and exotic plant species? We predict that parrots will interact similarly with both native and exotic plants[22]. However, their primary reliance on non-endozoochorous dispersal mechanisms[40] will enable them to integrate plant species, particularly those with non-drupaceous fruits that were not dispersed by birds in the pre-invasion network, into the mutualistic subnetwork.

## Results

During the study period, we accumulated approximately 288 h of observations, tracking the fate of 24,561 fruits from 576 individual plants. This effort resulted in the documentation of 243 unique interactions involving 24 bird, 13 native, and 21 exotic plant species with sampling effort sufficiently exhaustive to ensure that the majority of interacting species were reliably registered (Supplementary Methods 2 Figs. 2 and 3). Note that, for simplicity, we treated three plant genera, each comprising two morphologically similar species, as single species in our analyses. Among the fruits monitored across all bird species, 19% were defleshed, 5% were wasted and remained in the ground without being consumed during the focal sampling, 44% were predated and 32% were dispersed by endozoochory (78%), epizoochory (2%) or stomatochory (20%) (Supplementary Methods 3 Tables S1, 2 and 3). This information was used to build an antagonistic-mutualistic multilayer network (hereafter, invasion network). By removing the parakeets and all associated interactions and species that only interacted with them or were facilitated by them (Supplementary Table 4), we simulated the most likely scenario prior to the arrival of the invaders (hereafter, pre-invasion network) for comparison.

What role do invasive parakeets play in the multilayer network and its mutualistic and antagonistic subnetworks?

Invasive parakeets play central, transformative roles in the multilayer network and its mutualistic and antagonistic subnetworks, primarily by introducing novel interactions and altering network topology. The invasion network incorporated 88 novel antagonistic ($N = 41$) and mutualistic ($N = 47$) interactions exclusive to parakeets, many involving plants that were not previously consumed by the recipient bird community, broadening the plant richness in the invasion compared to the pre-invasion network (Fig. 1). Additionally, parakeets also facilitated 20 novel interactions through food wasting. In 71% of cases, parakeet-induced food waste temporarily facilitated access for other bird species to unripe seeds that lacked sufficient protection to survive gut passage, thereby increasing seed predation for plants that might otherwise have been dispersed. In 24% of cases, particularly with hard fruits and seeds, parakeets facilitated their consumption by birds that were unable to access these resources on their own (obligate facilitations) (Supplementary Table 4).

The inclusion of parakeets altered network modularity by redistributing species across modules while leaving the modularity values similar (Table 1; Fig. 2). Only 33% of the species pairs that interacted within the same module in the pre-invasion network remained together in the invasion network. Given that the parakeets directly or indirectly shifted the balance of interactions in the system, increasing antagonistic interactions at the expense of mutualistic ones (Fig. 1), many species that previously formed predominantly mutualistic modules began to combine within modules featuring mixed interactions (Fig. 2). Furthermore, parakeets, particularly the rose-ringed parakeet, increased the nestedness of the multilayer network and its mutualistic subnetwork beyond what would be expected by chance (Table 1 and Supplementary Fig. 2). Importantly, changes in network topology were primarily driven by species turnover induced by parakeets ($\beta_{ST}$: multilayer: 0.40; antagonistic subnetwork: 0.28; mutualistic subnetwork: 0.17) rather than by interaction rewiring ($\beta_{OS}$: multilayer: 0.01; antagonistic and mutualistic subnetworks: 0.00).

Parakeets assumed central roles in the multilayer invasion network and its subnetworks, both in terms of the number of plant species they interacted with (degree) and their connection to other highly connected species (eigenvector) (Fig. 3a). These central roles, previously held by subnetwork-specific native birds, such as *T. merula* in the mutualistic subnetwork and *P. domesticus* in the antagonistic subnetwork, were now dominated by parakeets. Despite the addition of new species, most plants retained their relative roles in the invasion networks (Fig. 3). Parakeets also acted as key connectors between antagonistic and mutualistic subnetworks. Most birds in the recipient community predominantly functioned as either antagonists or mutualists, with only a few individuals from certain species (*C. chloris*, *P. domesticus*, *P. pica*, and *S. borin*) engaged in both seed predation and dispersal. Following the entrance of parakeets, the occurrence of simultaneous

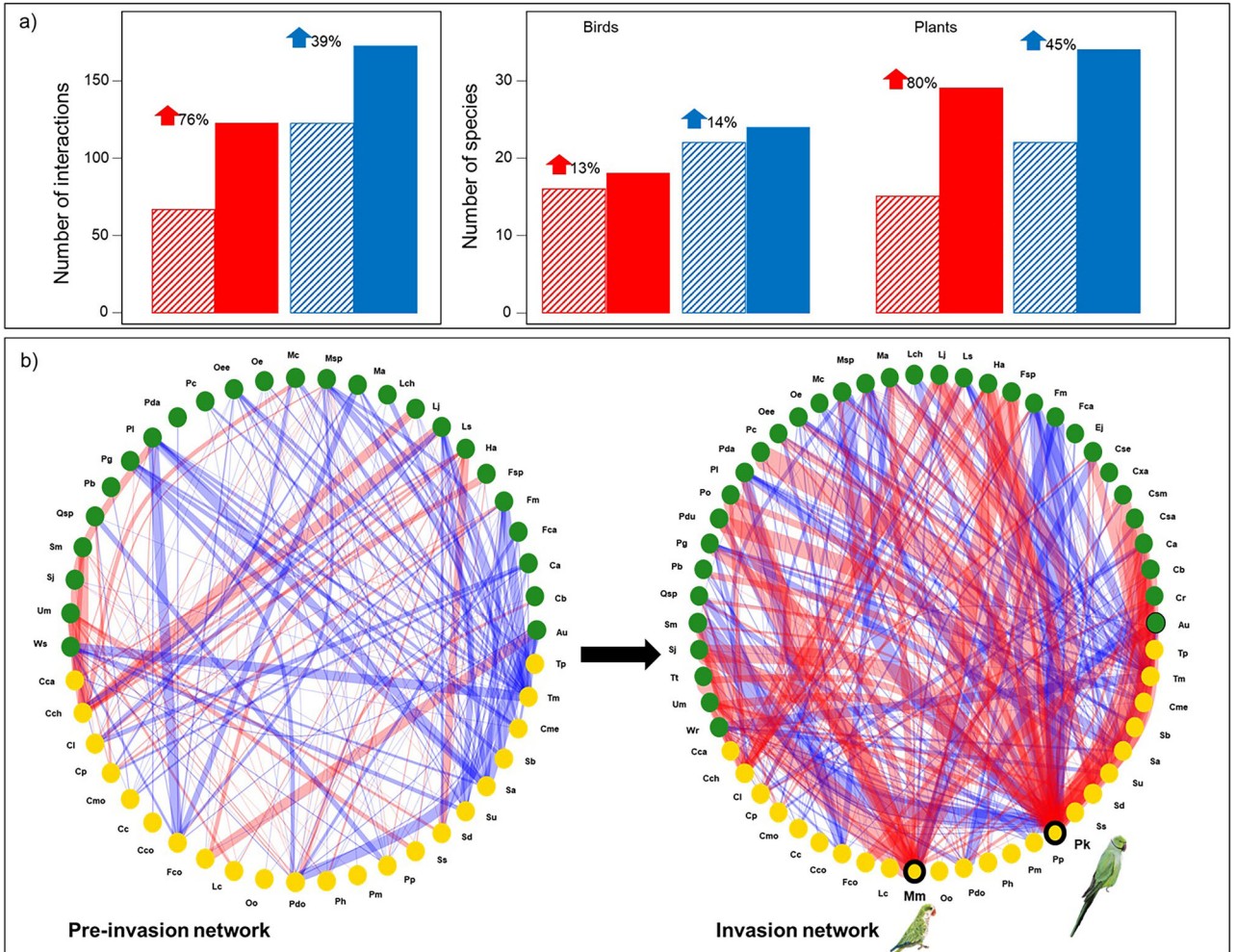

**Fig. 1 | Number of species and interactions and multilayer network structure before and after parakeet invasion. a** Number of interactions and bird and plant species included in the mutualistic (blue) and antagonistic (red) subnetworks before (pre-invasion; striped bars) and after (invasion; solid bars) the establishment of invasive parakeets in the system. **b** Multilayer networks showing the antagonistic (red lines) and mutualistic (blue lines) links between birds (yellow circles) and plants (green circles) in the pre-invasion and invasion networks. Line thickness is proportional to the weight of the interactions, representing the percentage of seeds of each plant species predated or dispersed by birds. Circles with black contours indicate invasive parakeets: the rose-ringed parakeet (*Psittacula krameri*, abbreviated Pk) and the monk parakeet (*Myiopsitta monachus*, abbreviated Mm). Plant species are abbreviated as follows: Au: *Arbutus unedo*, Ca: *Celtis australis*, Cb: *Catalpa bignonioides*, Cr: *Campsis radicans*, Csa: *Ceratonia siliqua*, Cse: *Cupressus sempervirens*, Csm: *Cercis siliquastrum*, Cxa: *Citrus × aurantium*, Ej: *Eriobotrya japonica*, Fca: *Ficus carica*, Fm: *F. microcarpa*, Fsp: *Fraxinus sp* (*Fraxinus angustifolia* and *F. excelsior*), Ha: *Helianthus annuus*, Lch: *Livistona chinensis*, Lj: *Ligustrum japonicum*, Ls: *Lantana strigocamara*, Ma: *Melia azedarach*, Mc: *Myrtus communis*, Msp: *Morus sp* (*Morus alba* and *M. nigra*), Oe: *Olea europaea*, Oee: *O. e. europaea*, Pb: *Pyrus bourgaeana*, Pc: *Phoenix canariensis*, Pda: *P. dactylifera*, Pdu: *Prunus dulcis*, Pg: *Punica granatum*, Pl: *Pistacia lentiscus*, Po: *Platycladus orientalis*, Qsp: *Quercus sp* (*Quercus ilex* and *Q. suber*), Sj: *Styphnolobium japonicum*, Sm: *Silybum marianum*, Tt: *Tipuana tipu*, Um: *Ulmus minor*, Wr: *Washingtonia robusta*. Birds species are abbreviated as follows: Cc: *Curruca communis*, Cme: *C. melanocephala*, Cca: *Carduelis carduelis*, Cch: *Chloris chloris*, Cco: *Cyanopica cooki*, Cl: *Columba livia var. dom.*, Cp: *C. palumbus*, Cmo: *Coloeus monedula*, Fco: *Fringilla coelebs*, Lc: *Linnaria cannabina*, Oo: *Oriolus oriolus*, Pdo: *Passer domesticus*, Ph: *P. hispaniolensis*, Pm: *P. montanus*, Pp: *Pica pica*, Sa: *Sylvia atricapilla*, Sb: *S. borin*, Sd: *Streptopelia decaocto*, Ss: *Serinus serinus*, Su: *Sturnus unicolor*, Tm: *Turdus merula*, Tp: *T. philomelos*.

predation and dispersal increased, with parakeets themselves emerging as critical linkers between the two subnetworks (Fig. 4a).

How do invasive parakeets interact with native and exotic plant species?

Although the composition of plant species in the networks did not significantly differ in origin before and after the entrance of parakeets (Contingency table for percentage of exotic plant species in the pre-invasion and invasion networks: $\chi^2 = 0.56$, $DF = 1$, $P = 0.4532$), the invasion network included greater number of plant species mainly due to the addition of exotics (the percentage exotic plants rose from 52% in the pre-invasion network to 62% in the invasion network). However, the native or exotic origin of plants did not influence their interactions with birds, as all plants exhibited stronger interactions weights in the invasion compared to the pre-

invasion network, independent on plant origin (Table 2; Fig. 4b). In the invasion network, both native and exotic plants interacted with more bird species overall, except in the case of antagonistic interactions (Table 2; Fig. 4b). Notably, although birds in the recipient community maintained the strongest mutualistic interactions, parakeets -particularly the rose-ringed parakeet- significantly increased epizoochory and, especially, stomatochory, two dispersal mechanisms rare in the recipient bird community studied (Table 2; Fig. 4b, c). Epizoochory, which was nearly absent in the pre-invasion network (only one case of *Sturnus unicolor* dispersing *Morus sp* seeds via epizoochory out of 5833 fruits dispersed), increased to 2% in the invasion network (152 out of 7918 fruits dispersed). Similarly, the proportion of fruits with intact seeds dispersed by stomatochory rose from 10% in the pre-invasion network to 20% in the invasion network (600 and 1595

**Table 1 | Network parameters of pre-invasion and invasion multilayer networks**

| All plant-bird interactions | Pre-invasion | | Invasion | |
|---|---|---|---|---|
| | **Nestedness** | **Modularity** | **Nestedness** | **Modularity** |
| Multilayer network | 52.24 | **0.41** | **58.27** | 0.35 |
| | (48.92–62.04) | (0.15–0.19) | (47.37–58.06) | (0.09–0.12) |
| Antagonistic subnetwork | 60.00 | **0.42** | 60.09 | **0.37** |
| | (44.13–63.19) | (0.16–0.23) | (50.39–65.04) | (0.09–0.13) |
| Mutualistic subnetwork | 55.47 | **0.37** | **65.82** | **0.38** |
| | (46.99–61.3) | (0.15–0.20) | (47.96–61.05) | (0.13–0.17) |

Parameters are shown for the multilayer networks and their two constituent layers (antagonistic and mutualistic subnetworks) before (pre-invasion) and after (invasion) the establishment of invasive parakeets. Observed values that significantly differ from those expected under the null models are shown in bold.

fruits dispersed by stomatochory in the pre-invasion and invasion network, respectively). This shift led to a substantial increase in the number of plant species dispersed by stomatochory in the invasion compared to the pre-invasion network (Fig. 4b). Importantly, all nine plant species newly integrated into the invasion network were dispersed by parakeets, predominantly throughout stomatochory and, to a lesser extent, epizoochory. However, since both native and invasive bird species interact with exotic plant species, the most common dispersal mechanism for these plants remains endozoochory overall (81% and 76% of fruits dispersed via endozoochory in exotic and native plants, respectively; $\chi^2 = 20.48$, $DF = 1$, $P < 0.0001$).

## Discussion

Parrots are highly popular pets, and their extensive trade, coupled with frequent escapes and intentional releases, has resulted in the establishment of numerous exotic populations worldwide[46]. Among these, rose-ringed and monk parakeets stand out for their invasive potential[35], causing considerable ecological and socioeconomic impacts[43]. This study sheds light on a lesser-known aspect of their invasion: their effects on interaction networks within recipient communities. Through detailed monitoring conducted over a full annual cycle of bird–plant interactions in urban and peri-urban environments–common establishment zones for these species worldwide-we reveal that invasive parakeets, mainly the rose-ringed parakeet, can alter network structural properties, such as modularity and nestedness. By functioning as both antagonists and mutualists, parakeets enhance connectivity between previously isolated subnetworks, while their unique traits create novel interactions between species and promote dispersal mechanisms uncommon in recipient communities. These changes can benefit some native plant and bird species, but they also risk spreading exotic plants, potentially triggering secondary invasions.

In the pre-invasion network, as in most Mediterranean systems, plant–bird interactions were predominantly mutualistic, driven mainly by the endozoochorous dispersal of fleshy fruits, and, to a lesser extent, antagonistic[47,48]. The entrance of invasive parakeets increased the number of plant species and interactions in the network, especially within the antagonistic subnetwork, by incorporating species not previously involved in interactions. Parakeet's powerful beaks and physiological adaptations for detoxifying secondary metabolites in fruits and seeds[38] allow them to predate seeds from a wide variety of plants, including those previously unused by the local avian community. In some occasions, parakeets did not fully destroy the fruits or seeds they consumed, and interactions that began as antagonistic often transitioned into effective dispersal when parrots discarded intact seeds far from the mother plant (dispersal via stomatochory and, to a lesser extent, epizoochory). Though there is growing evidence of this mutualism-antagonism continuum in other species[49,50], parakeets interact as both antagonists and mutualists with a broadest range of plants in our study area, enhancing connectivity between network layers. A notable feature of parrot foraging is their extensive food-wasting behavior, which exceeds that of most other species. Food waste provides access to plants for birds that might otherwise be unable to reach their fruits or seeds (obligate facilitation) or could only do so during specific phenological stages (temporary facilitation)[44], fostering novel interactions. While not directly addressed in this study, parrot-induced food waste also supports secondary dispersals by other animals, such as mammals or ants[44,51,52], expanding the diversity of seed dispersers -with varied home ranges, dispersal strategies or retention times- and therefore enhancing dispersal success and plant colonization potential[52,53].

Although the networks including invasive bird species retained the same modular structure as the pre-invasion networks, the distribution of species among modules changed. Parakeets –particularly the ringed-neck parakeet- also increased nestedness within the mutualistic subnetwork, ultimately affecting the overall configuration of the multilayer invasion network. The topology of plant-bird interactions networks is shaped by different ecological processes[54–56], including morphological limitations[57,58] and, maybe most importantly, species abundances[8,59]. While we lack comprehensive data on the relative abundance of all bird species in the study community to explicitly assess the relevance of neutral processes in driving nestedness[60], the large populations of both invasive parakeets may help explain their strong influence on network structure. Since their introduction in Seville in the 1990s, their numbers have steadily increased[61], in line with broader European trends[62,63]. However, their population sizes differ locally -likely due to distinct invasion histories[61,64] and, together with their broad but different diets and foraging strategies (with rose-ringed parakeets being more arboreal) may account for their differing roles in shaping network architecture and functionality.

Birds disperse seeds through multiple mechanisms[39,65], though most mutualistic interactions in the pre-invasion network occurred via endozoochory. Parrots' role as endozoochorous dispersers has been largely underestimated, but viable small seeds have been found in their feces[66,67]. Additionally, parrots can disperse seeds by carrying them in their beaks or legs (stomatochory[41]) or attached to their bodies (epizoochory[10]), making them keystone dispersers for a wide variety of plants[39]. The entrance of parakeets in the system has increased the prevalence of these external dispersal mechanisms, which were almost absent among native birds. Unlike others examples where seeds attach passively to the animal's body with no benefit to the transporter[68], epizoochory in parrots is a mutualism, as seeds adhere to the bird's face or beak while individuals are feeding on fruits with sticky pulp[10]. Stomatochory, on the other hand, was formerly limited to species like corvids, thrushes and starlings in the study system[20], but became more frequent after the invasion by parakeets, especially for large-seeded fleshy and dry-fruited plants. Although not included in our analyses due to their low numbers, two additional parrot species are currently present in the study area. Given that all parrots share similar feeding behaviors, it is likely that these species contribute similarly to the dynamics described for the two focal species. However, as we observed differences between the rose-ringed and monk parakeets, it is also possible that such differences exist with these other species, and therefore, their roles should be investigated in future studies.

Most invasive parrot populations inhabit urban areas[35,62], where, according to our results, can play an important role as seed predators and dispersers[69], sometimes reinforcing or even replacing the ecological functions of native species occurring at lower densities[9,70]. However, both parakeets are currently expanding their ranges, increasingly colonizing peri-urban and rural landscapes[64,71], and extending their ecological influence beyond city boundaries. Indeed, we frequently observed these parakeets foraging outside urban cores, particularly in rural gardens and scattered dwellings where many exotic plants, commonly cultivated for ornamental use in public and private spaces[72], are abundant. Horticulture is a major pathway for the introduction of exotic plants, some of which can become invasive when paired with effective seed dispersers[73,74]. Parakeet's foraging behavior, combined with their high mobility, allows them to disperse seeds of native and exotic plants[22] from human-dominated areas into natural or semi-natural habitats, facilitating plant range expansions and generating

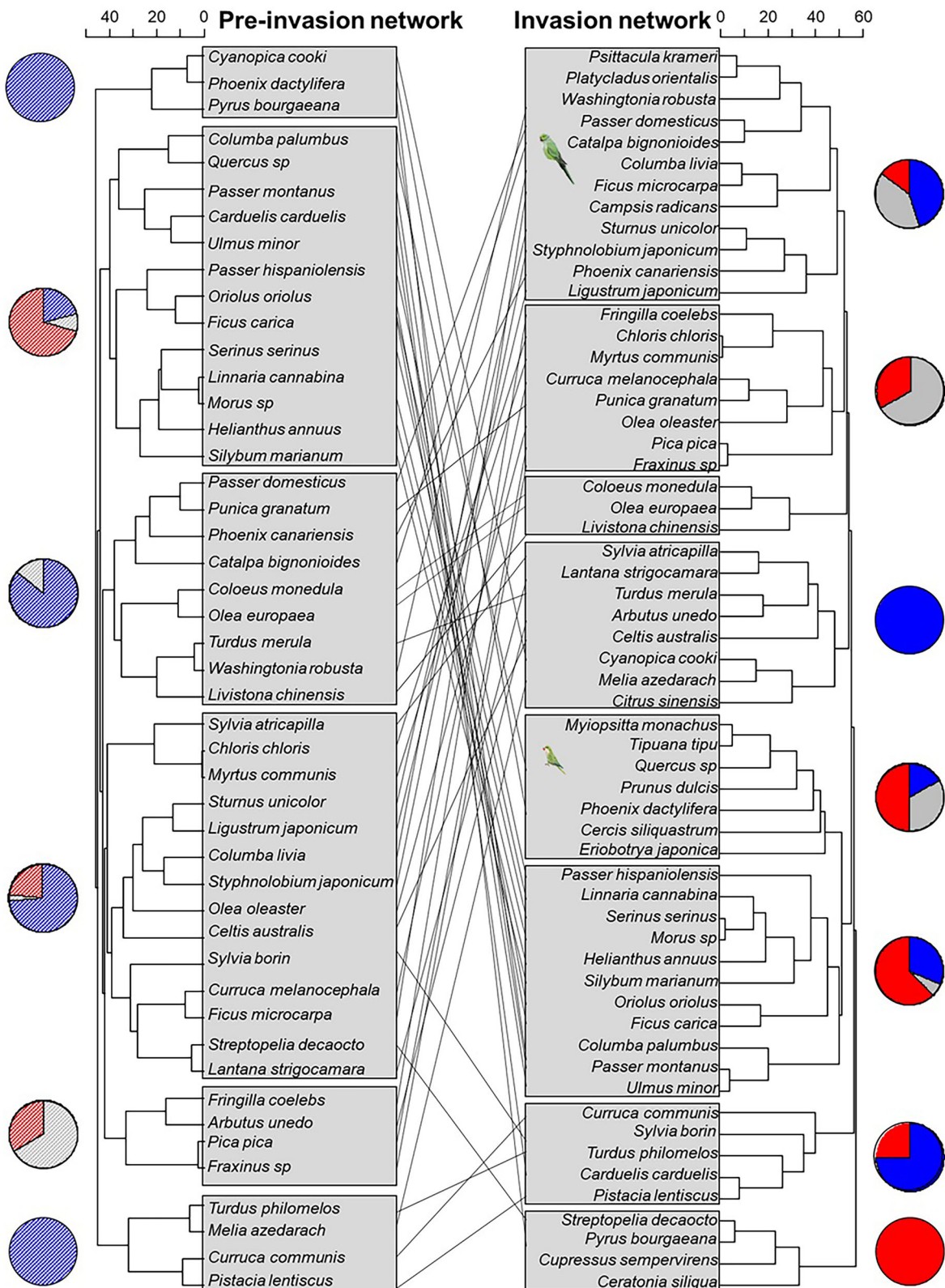

**Fig. 2 | Distribution of plant and bird species across modules in the pre-invasion and invasion multilayer networks.** Modules are shown for the pre-invasion (right) and invasion (left) networks. Lines connect species that change modules after the incorporation of parrots in the system. Pies chart slices represent the proportion of species within each module engaged in mutualistic (blue), antagonistic (red) or dual (i.e., mutualistic and antagonistic; gray) interactions in pre-invasion (striped pies) and invasion (solid pies) networks. The horizontal axis of the dendrogram is expressed in dissimilarity units produced by the hierarchical clustering of modules using the edge-betweenness community detection algorithm (*igraph*), where larger values represent greater structural dissimilarity between species clusters.

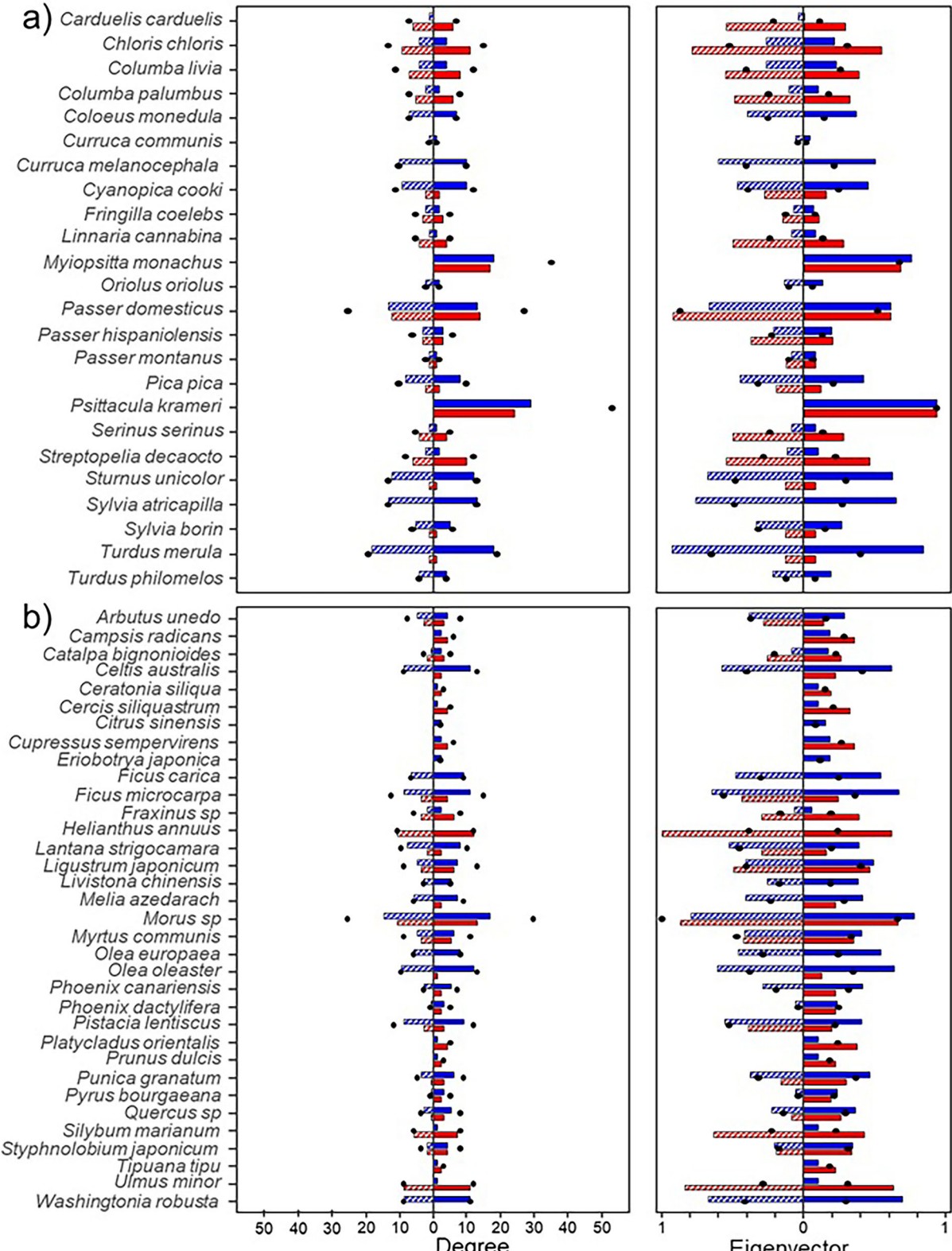

**Fig. 3 | Centrality of bird and plant species in pre-invasion and invasion multilayer networks.** Centrality metrics obtained for **a** bird and **b** plant species in the pre-invasion (striped bars) and invasion (solid bars) networks. Black dots indicate centrality values in multilayer networks, blue bars in the mutualistic subnetwork and red bars in the antagonistic subnetwork. Centrality was quantified by degree (i.e., the number of interacting species; left) and eigenvector centrality (i.e., connectivity to highly connected species; right).

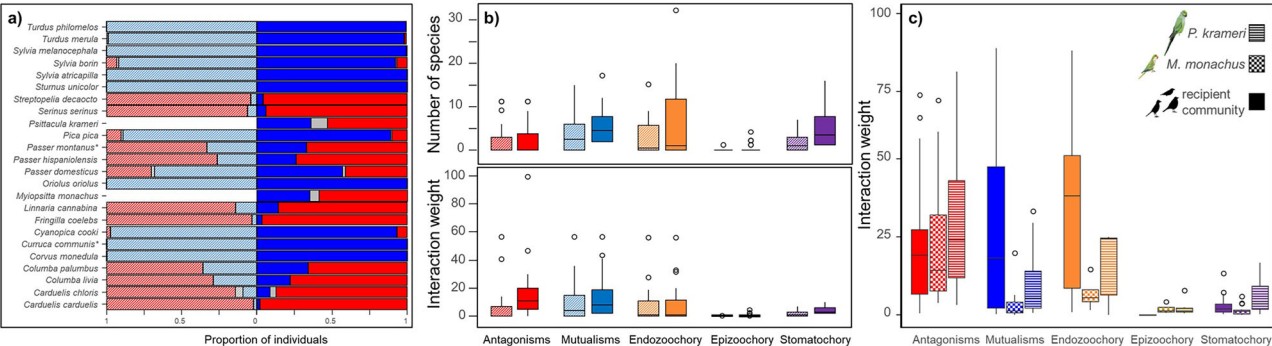

**Fig. 4 | Changes in bird roles and interactions before and after the establishment of parakeets. a** Roles of birds as seed predators and dispersers before and after parakeet invasion. **a** Proportion of individuals within each bird species engaged in seed predation (red bars), dispersal (blue bars) or both behaviors simultaneously (gray bars) during focal sampling periods (up to 5 min). Pre-invasion interactions are represented by striped bars and invasion interactions by solid bars. **b** Boxplots showing the mean number of plant species and mean interaction weights for birds in pre-invasion (striped bars) and invasion (solid bars) networks. Interactions include seed predation and dispersal via endozoochory, epizoochory and stomatochory. **c** Boxplots of mean interaction weights for parakeets compared with all birds of the recipient community. In panels **b** and **c**, boxplots indicate the median (central line), interquartile range (box), minimum and maximum values within 1.5 times the interquartile range (whiskers) and outliers (points).

**Table 2 | Statistical models comparing species richness and interaction patterns**

| a | N bird species | | | Interaction weight | | | b | Interaction weight | | |
|---|---|---|---|---|---|---|---|---|---|---|
| **Antagonisms** | $\chi^2$ | DF | P | $\chi^2$ | DF | P | **Antagonisms** | $\chi^2$ | DF | P |
| Plant origin | 1.57 | 1 | 0.2108 | 1.19 | 1 | 0.2753 | Bird | 3.37 | 2 | 0.1855 |
| Network | 0.06 | 1 | 0.7991 | 34.38 | 1 | <0.0001 | Plant origin | 0.93 | 1 | 0.3337 |
| **Mutualisms** | $\chi^2$ | DF | P | $\chi^2$ | DF | P | **Mutualisms** | $\chi^2$ | DF | P |
| Plant origin | 1.00 | 1 | 0.3184 | 5.05 | 1 | 0.0247 | Bird | 20.50 | 2 | <0.0001 |
| Network | 8.38 | 1 | 0.0038 | 12.79 | 1 | 0.00035 | Plant origin | 1.35 | 1 | 0.2448 |
| **Endozoochory** | $\chi^2$ | DF | P | $\chi^2$ | DF | P | **Endozoochory** | $\chi^2$ | DF | P |
| Plant origin | 0.20 | 1 | 0.6574 | 0.72 | 1 | 0.3954 | Bird | 3.80 | 2 | 0.1493 |
| Network | 36.36 | 1 | <0.0001 | 4.16 | 1 | 0.0414 | Plant origin | 2.85 | 1 | 0.0912 |
| **Epizoochory** | $\chi^2$ | DF | P | $\chi^2$ | DF | P | **Epizoochory** | $\chi^2$ | DF | P |
| Plant origin | 0.10 | 1 | 0.7525 | 0.06 | 1 | 0.8105 | Bird | 2.16 | 2 | 0.3392 |
| Network | 5.70 | 1 | 0.0170 | 5.32 | 1 | 0.0211 | Plant origin | 6.58 | 1 | 0.0103 |
| **Stomatochory** | $\chi^2$ | DF | P | $\chi^2$ | DF | P | **Stomatochory** | $\chi^2$ | DF | P |
| Plant origin | 0.22 | 1 | 0.6367 | 0.65 | 1 | 0.4189 | Bird | 15.63 | 2 | 0.0004 |
| Network | 45.03 | 1 | <0.0001 | 34.17 | 1 | <0.0001 | Plant origin | 0.04 | 1 | 0.8487 |

**a** Models testing the effects of plant origin (native vs exotic) and network type (pre-invasion vs invasion) on the number of bird species (N bird species) and interaction weight. **b** Models testing the effect of bird taxa (rose-ringed parakeet *Psittacula krameri*, monk parakeet *Myiopsitta monachus* and recipient community birds) on the weight of antagonistic and mutualistic interactions (Antagonisms and Mutualisms, respectively) and on the different dispersal mechanisms (Endozoochory, Epizoochory and Stomatochory). Plant origin was obtained from the literature (see Supplementary Tables 1 and 2).

novel ecological dynamics. Although the total number of exotic plant species in the network did not increase notably following parakeet establishment, the composition and functional nature of those interactions changed. Many exotic plants in our system are widely cultivated and some are listed as invasive in other parts of the Mediterranean or North America[75,76]. While this species may have remained confined to managed urban environments thus far, parakeet-mediated dispersal could shift that balance. This process exemplifies the potential for secondary invasion[77], a relatively understudied phenomenon where the invasion success of an exotic species is entirely dependent on the presence and influence of another, which itself can successfully establish without the assistance of another exotic. Despite their broad dietary range, parakeets are not reliant on any single plant species for establishment or spread. Still, they can act as key dispersal agents for the exotic plants they interact with, either directly or by facilitating dispersal by other animals. In fact, some exotic plants in the invasion network were exclusively dispersed by parakeets, predominantly via stomatochory. Seeds of exotic plants with dry fruits may disperse farther through stomatochory than previously assumed, especially if non-zoochorous (e.g., anemochory[78]; but see refs. [79,80].) or less mobile animals (e.g., rodents or insects[23,81]) are involved. Additionally, because many dry fruits contain multiple seeds, stomatochory often results in the simultaneous movement of multiple propagules, increasing seed survival rates even after partial predation[20,41,69,82]. Visual tracking of birds in our study system and others suggests that seeds dispersed by stomatochory can travel substantial distances, often up to several hundred meters[22,41,42,79]. While more studies are needed to confirm the long-term outcomes, these novel interactions, driven by the range expansion of invasive parakeets, highlight an emerging ecological dynamic. Even if secondary invasions are not yet widespread, the introduction of new dispersal pathways—especially involving exotic species previously unassisted by native birds—is ecologically relevant. Given the parakeet's dispersal capacity and adaptability, they may become influential agents shaping future plant community dynamics across urban–rural gradients.

Understanding how invasive species alter ecological networks is inherently complex, as it requires comparisons with scenarios in which those species are absent. Ideally, this would involve analyzing independent ecological networks along invasion gradients or comparing interactions

before and after the establishment of the invaders. However, such situations are rare and often confounded by other ecological factors, making this approach logistically unfeasible. To overcome these problems, researchers often rely on alternative methods, such as network meta-analyses to identify overarching patterns[18], species-level interaction analysis[83], or simulations using null models to construct hypothetical baseline scenarios[67]. In this study, we simulate a pre-invasion network by excluding interactions involving invasive species and their associated facilitative processes, using data collected at a single site and time point to minimize variability in species abundances. Although previous studies have shown that parakeets can reduce feeding rates among native birds at feeding stations[84,85], this interference tends to decline over time as natives habituate to their presence[84]. In our study area, where parakeets have established since the 1990s, there is no direct or indirect evidence that they have displaced or reduced the abundances of the native species included in our network. Therefore, our approach would offer a practical starting point for investigating the effects of invasive species, while future studies collect data from nearby uninvaded areas to more directly evaluate their impact on network functioning. Moreover, we also recognize that our single-year study design cannot capture inter-annual variability in climate, resource availability or species abundances, all factors that may influence plant-bird interactions and, by extension, network dynamics. Although our dataset provides exceptional temporal resolution, capturing a complete phenological cycle of plants and birds, it may still overlook important sources of ecological variation, including fruiting phenology shifts, extreme climatic events (e.g., droughts, frosts), changes in bird foraging behaviors or rare phenomena like masting or irruptive migrations[39]. Future multi-year studies will be crucial to disentangle transient environmental effects from the persistent impacts of invasive parakeets on ecological networks, especially in the context of accelerating global change. To truly understand the extent of these impacts, it is also important to evaluate the effectiveness of seed dispersal in the urban environments these species typically inhabit. While previous research confirms that parrots are legitimate seed dispersers[39–42,66,67,69], dispersal in human-dominated environments or involving exotic plant species, which may sometimes be sterile, does not necessarily lead to successful germination, growth and survival of seedlings. Although beyond the scope of our study, understanding whether parakeet-mediated seed dispersal leads to plant germination and growth is critical to assessing the potential ecological risks or benefits posed by these invasive species.

Biological invasions can profoundly alter species interactions[16,86], making it essential to understand how invasive species integrate into communities to predict their impacts. These impacts are often linked to the ecological roles invasive species play in their native ranges, as similar mechanisms may govern these roles in novel environments[87]. Parrots, belonging to one of the most diverse[88,89] and abundant[90] avian orders, have historically been understudied in terms of their ecological contributions, with a predominant focus on their role as seed predators[90,91] rather than effective seed dispersers[10,20,39,40]. Consequently, research on their impacts in invasive areas has primarily emphasized vegetation and crop damage[43]. This perspective has also excluded parrots from studies on frugivory networks[87,92,93] (but see ref. 94), despite their dual role as seed predators and dispersers in their native ecosystems is essential for establishing intricate ecological networks that are stable and resilient to disturbances[26]. In this sense, it is crucial to re-evaluate parrot's contributions to tropical ecosystems, particularly in light of the potential consequences of their population declines. Furthermore, as demonstrated in this study, the ecological impacts of invasive parrots should be extended beyond competition with native species[43,95] to include their capacity to restructure interaction networks to resemble those of their native ecosystems and facilitate the spread of exotic plants. These impacts can greatly modify the functionality of recipient ecosystems and are likely to become more serious as invasive populations continue to increase, both numerically and geographically. It is important to acknowledge that the two parakeets studied here have distinct effects on ecological networks, suggesting that their roles may not be redundant. Further research should aim to clarify whether these differences are the result of functional disparities between the species or simply a reflection of their varying population sizes[61] and invasion histories[64]. These factors merit deeper investigation to understand the underlying causes of their ecological impacts.

## Methods

### Study area and species

The study was conducted between April 2022 and May 2023 in the city of Seville (southwestern Spain) and its metropolitan belt (Supplementary Fig. 2). This area (approximately 735 km$^2$) has a Mediterranean climate and includes a patchwork of urban landscapes, parks with ornamental trees and rural zones with natural habitats and intensive agriculture (mainly olives, cereals, oranges, and sunflowers) along the Guadalquivir, Guadiamar and Guadaíra rivers. Bird species inhabiting both natural and humanized ecosystems in the study area are primarily frugivorous (families Columbidae: *Columba palumbus*, Corvidae: *Coloeus monedula*, *Cyanopica cooki* and *Pica pica*, Oriolidae: *Oriolus oriolus*, Sturnidae: *Sturnus unicolor*, Sylviidae: *Sylvia atricapilla*, *S. borin*, *Curruca communis* and *C. melanocephala* and Turdidae: *Turdus merula* and *T. philomelos*) and granivorous (families Passeridae: *Passer domesticus*, *P. hispaniolensis* and *P. montanus* and Fringillidae: *Linaria cannabina*, *Carduelis carduelis*, *Chloris chloris*, *Fringilla coelebs* and *Serinus serinus*). Most of these birds are resident species, though some, such as *S. borin*, *O.oriolus* or *T. philomelos*, are present in high abundance only during certain months of the year[96,97]. The native bird community also included a feral species (*Columba livia var. dom.*) and a natural invader (*Streptopelia decaocto*) and was augmented in the early 1990s by two invasive parrots, the rose-ringed and the monk parakeets, whose populations in 2024 were estimated at approximately 8000 and 3000 individuals, respectively[61]. Two other parrot species, the Senegal parrot (*Poicephalus senegalus*) and the blue-crowned parakeet (*Thectocercus acuticaudatus*), are also present but in very low numbers (1 and 6 individuals, respectively) and were excluded from analyses.

### Data collection

Plant diversity in the study area was too vast to monitor comprehensively. We therefore conducted preliminary work to identify the main plant species utilized by birds for feeding. These selected species formed the basis for constructing the ecological networks and included native Mediterranean and exotic ornamental species with arboreal and shrub-like growth habits. Herbaceous plants were excluded due to the challenges in standardizing their monitoring compared to trees and shrubs. However, exceptions were made for sunflowers (*Helianthus annuus*) and blessed thistles (*Silybum marianum*) due to their size (approximately 2 m height) and the fact that they were heavily consumed by several bird species. This focused approach allowed us to concentrate on 34 native and exotic plant species from 18 families, representing a wide array of fruit morphologies (dry, drupaceous, and berry-like fruits; Supplementary Table 1 and 2).

We recorded bird-fruit interactions weekly over the course of a full annual cycle. To capture the spatial distribution of bird species while considering site accessibility, individual plants were selected by stratifying the study area into distinct sectors, including all major parks, gardens, and adjacent rural areas with more natural vegetation (Supplementary Fig. 2). Based on previous fieldwork, we knew that not all individual plants would be visited by birds. Therefore, we surveyed each area by walking through it and identifying target plants; once bird foraging activity was detected, we began recording interactions. We monitored as many individuals of each plant species as possible to maximize detection of interactions. To assess the adequacy of our sampling effort in capturing plant-animal interactions, we generated species accumulation curves for each plant species using the *BiodiversityR* 2.16.1 package[98] (Supplementary Fig. 2). In some cases, particularly for plant species that attracted very few bird species, we conducted additional observations to ensure that the lack of stabilization in interaction accumulation curves was not due to insufficient sampling. For these species, we even retained the data when the patterns aligned with ecological expectations (e.g., fruits not typically consumed by native birds). To

minimize pseudoreplication, each plant was GPS-recorded and observed for 30 min at each stage of its fruit phenology (unripe and ripe), provided that bird foraging activity was detected. During these sessions, foraging individuals were observed for 5 min, and all interactions with fruits were recorded. Interactions were classified as seed predation, fruit defleshing or seed dispersal by endozoochory, epizoochory or stomatochory based on the bird's actions and the fate of the seeds of each fruit. When birds completely destroyed the seeds, the interaction was classified as seed predation. If birds consumed only the pulp of the fruit, it was considered fruit defleshing. For acorns (seeds from trees of the genus *Quercus*), partial consumption of cotyledons (less than 60%) does not damage the embryo and facilitates germination[99,100]. This is analogous to pulp removal in fleshy fruits[101,102], so these observations were classified as fruit defleshing. Interactions where birds only defleshed fruits were considered as commensalists and excluded from network analyses as it was uncertain whether this action benefits or harms germination, even though it clearly benefits the birds. Seeds that were swallowed whole were generally considered dispersed by endozoochory unless damaged while passing through the bird's digestive tract. Such instances, confirmed through fecal and regurgitate analysis, included: 1) ingestion of unripe fruits with unprotected seeds[66], 2) ingestion of ripe fruits with small seeds ( < 5 mm) and soft coats (seed hardness = 1; Supplementary Table 1) by pigeons and doves[103,104] and 3) ingestion of acorns by the wood pigeon (*Columba palumbus*), whose gizzard completely crushed them[105]. In some instances, especially with berry-like fruits, some seeds were consumed with the pulp, and the interaction was therefore classified as endozoochory. When birds carried viable seeds away from the mother plant, the interaction was classified as stomatochory if the transport was active, involving the bird carrying intact seeds in the beaks or feet, or as epizoochory if seeds were passively transported attached to the bird's body. To confirm stomatochory as an effective seed dispersal mechanism, we visually tracked the birds whenever possible and examined the fate of the seeds in situ (i.e., whether they were predated, swallowed or wasted). In some cases, birds handled fruits without consuming either the pulp or the seeds, subsequently discarding them on the ground. These observations, classified as food wasting[44], were treated as neutral and also excluded from analyses unless the seeds were subsequently predated or dispersed by another bird species during the observation period[45]. Note that both wasted and defleshed fruits were considered in the overall count of handled fruits (see below). Observations were made by 2–3 people using binoculars and telescopes, maintaining a minimum distance of 15–50 m from the focal plant. Monitoring occurred during the first three hours after sunrise and the last three hours before sunset, coinciding with the peak of bird foraging activity, while avoiding suboptimal weather conditions, such as strong wind or rain. Due to high morphological similarity and frequent hybridization, interactions involving *Quercus ilex* and *Q. suber*, *Fraxinus excelsior* and *F. angustifolia*, and *Morus alba* and *M. nigra* were grouped at the genus level (*Quercus*, *Fraxinus* and *Morus*, respectively). However, interactions with *Olea europaea* and *O. e. europaea*, *Phoenix canariensis* and *P. dactylifera* and *Ficus carica* and *F. microcarpa* were analyzed separately, as these species exhibit marked differences in fruit size. The study is observational only and no live animals were collected.

## Plant-bird interaction networks

We constructed a quantitative multilayer network, with separate layers representing antagonistic and mutualistic interactions (i.e., mutualistic and antagonistic subnetworks; Supplementary Table 1). Due to the lack of data from before the establishment of parakeets in the study area, we created an additional multilayer network that excluded all parakeet interactions as well as those of local birds facilitated by parakeets (Supplementary Table 4). The first network represents the invaded community (invasion network), while the second simulates the most plausible pre-invasion scenario (pre-invasion network). Although this approach is not without limitations, it enables meaningful comparisons while maintaining the native species' abundances unchanged–a scenario that would not have been feasible had a different study area been used as a reference for the pre-invasion condition. To

quantify interaction strength (interaction weight, *wij*) for a link connecting a bird species *i* and a plant species *j*, we used the number of seeds involved in mutualistic (i.e., dispersed) or antagonistic (i.e., predated) interactions as a percentage of the total number of seeds predated, dispersed, defleshed or wasted by all bird interacting with a given plant species, including the two parakeets. The mean number of seeds per fruit was obtained from the literature (Supplementary Table 2). This approach provided the best approximation for assessing the relative contribution of each bird species to the plant community, regardless of the total number of seeds consumed.

## Statistics and reproducibility

We analyzed changes associated with parakeets in the topology of the multilayer network and its mutualistic and antagonistic subnetworks by measuring their nestedness and modularity before and after the invasion. Nestedness, which reflects the presence of hierarchies in species interactions, was calculated using the NODF metric (Nestedness metric based on Overlap and Decreasing Fill[106]). NODF ranges from 0, indicating a completely modular structure, to 100, representing a perfectly nested structure. To identify discrete clusters of species that are densely interconnected but weakly linked to other clusters, we calculated the weighted modularity Q of the networks[107]. To assess whether the observed nestedness and modularity significantly differed from random expectations, we randomized the interaction matrices 1000 times using null models that preserved the observed connectance (i.e., the ratio of observed to possible interactions[60]) of the networks in both invasion and pre-invasion scenarios. The observed metrics were then compared with the frequency distribution generated by the null models to determine whether they were significantly higher or lower than expected by chance. To evaluate whether parakeets alter network topology by modifying existing interspecific interactions (rewiring, $\beta_{OS}$) or changing species composition (species turnover, $\beta_{ST}$), we calculated the dissimilarity in subnetwork structure between the pre-invasion and invasion scenarios[108].

To evaluate the role of parakeets in the networks, we calculated the degree (i.e., the total number of connections a species has with others) and the eigenvector centrality (i.e., number and influence of connections a species has, including the connectivity of its linked species) for both birds and plants in the pre-invasion and invasion networks and their mutualistic and antagonistic subnetworks. To minimize biases introduced by unmeasured factors, such as species abundance, which can influence interaction quantification[87], we adopted a qualitative approach. This method avoids depicting context-specific situations overly depend on local species abundances, offering a more generalizable perspective on species interactions across scenarios. We also examined how the dual role of parrots as both seed predators and dispersers influenced the strength of connections between mutualistic and antagonistic subnetworks. Specifically, we calculated the proportion of individuals within each species that engaged in seed predation or dispersal (intralayer connections) or performed both behaviors simultaneously (interlayer connections) during focal sampling periods (up to 5 min) in both the pre-invasion and invasion networks[109].

To understand how parakeets interacted with plants based on their origin, we used Generalized Linear Models to compare the number of bird species (Poisson error distribution, log link function) interacting with each plant and the weight (log-transformed; normal error distribution, identity link function) of these interactions –covering both antagonisms and mutualisms and including the specific dispersal mechanisms involved– between native and exotic plants in the pre-invasion and invasion network (fixed factors). The number of exotic and native plants in networks with and without parakeets were compared using Chi-squared contingency tables. Focusing specifically on mutualistic interactions, we further evaluated whether parakeets dispersed native and exotic plants differently and whether their dispersal mechanisms differed from those used by the recipient bird community. For this, we compared the strength of mutualistic interactions (*wij*; endozoochory, epizoochory and stomatochory) involving the two parakeets with those involving all birds in the recipient community collectively (fixed factor). When necessary, plant species were included as a

random term in models to minimize pseudo-replication. All analyses were conducted in R 4.3.1[110], using the packages *bipartite* 2.18[111], *igraph* 1.5.1[112] and *glmmTMB* 1.1.10[113].

## Reporting summary

Further information on research design is available in the Nature Portfolio Reporting Summary linked to this article.

## Data availability

All data generated or analyzed during this study are included in this published article (and its supplementary information files).

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

## Acknowledgements
We would like to thank Lucía Morillo for her help in the field work. This research was financially supported by the Plan Complementario I + D + i Biodiversidad (PCBIO BIOD22_00033_7_PPCB) funded by EU within the framework of the Recovery, Transformation and Resilience Plan (NextGenerationEU), the Regional Government of Andalucia and MICINN (European Regional Development Fund, SUMHAL, LIFEWATCH-2019-09-CSIC-13, POPE 2014-2020).

## Author contributions
J.I.P. performed fieldwork, analyzed data, wrote the paper, D.H.B. designed and performed fieldwork, wrote the paper, F.H. conceived the idea, designed, and performed fieldwork, wrote the paper, D.G.C. wrote the paper, J.L.T. conceived the idea, designed fieldwork, wrote the paper, M.C. conceived the idea, designed and performed fieldwork, analyzed data, wrote the paper.

## Competing interests
The authors declare no competing interests.
