## [Transparent Peer Review file · Communications Biology]

Multi-layer networks reveal changes in plant-bird interactions driven by invasive species

Corresponding Author: Dr Dailos Hernández-Brito

Version 0:

Reviewer comments:

Reviewer #1

(Remarks to the Author)

Thank you for the opportunity to review this interesting and well written paper on the ways that invasive parrots can disrupt and restructure plant-animal interactions. I think this paper will be of broad interest given that the focal species (two species of parakeets) have been introduced and established in dozens of countries across continents. More generally, understanding how invasive species change ecological processes such as seed predation and dispersal, including their potential to catalyze secondary invasions, is both ecologically interesting and critical to maintaining native biodiversity. Nonetheless, I do have the following concerns and suggestions for improvement.

Major Comments

My largest concern is that there are many aspects of the methodological approach that are unclear, limiting my ability to evaluate the findings, and the ability of others to replicate this study. For example, was the spatial scope of your study the entire city of Seville? How were the individual plants selected for observation? Were these plants/locations randomly selected, stratified across the city or in relation to the abundance of the invasive parakeets, etc.? What are your sample sizes – e.g., the number of individuals of each plant species observed? How did you select these sample sizes to ensure sufficient power to detect patterns? Finally, how did you track the fate of fruits/seeds and determine whether interactions were mutualistic or antagonistic? This seems extremely challenging given these birds are highly mobile and fast flying.

I would be careful to clarify that dispersal does not necessarily result in successful germination, growth and survival of seeds/seedlings. Particularly in urban systems, random dispersal vs dispersal into particular microclimates may be critical for plant recruitment. As such, I would be cautious in stating that parakeets could augment or replace functions performed by native species at lower densities.

Minor Comments

Abstract

Lines 17-18: Unless this is discouraged by the journal, I suggest adding a methods sentence that indicates where and how this question was addressed.

Line 19: Should read “Acting as both...”
Introduction

Line 62: Is this mutualism if the bird does not receive a benefit? Maybe clarify that this is mutualism only if the bird consumes some of the fruit but does not damage the seed?

Lines 78-82: I think it would be helpful to include a global map that shows the native and introduced ranges of your focal species, with a symbol to indicate your study site. This could go in supplemental materials if there is not room in the main text. This could help illustrate the potential widespread relevance of your study.

Lines 94-96: Is it correct that you only collected 1 year of data? I am confused as here you state data was collected over a full

annual cycle but later (Line 200) you refer annual monitoring, which suggests monitoring over multiple years. If the study was limited to one year that is understandable given the scope of the effort within this year, but I think it is important to discuss the potential limitations of this approach. Is there reason to believe that inter-annual variability in climate or other factors could impact bird and plant populations, phenology of foraging and fruiting, and species interactions? Also, were you able to look at the effect of seasons on species interactions within your study year?

Discussion

Lines 232-233: Can you speak further to any differences between the two parakeet species in regard to the interaction networks? This is referenced again on lines 316-318, but not discussed in any detail. In what ways do these two species have different ecological roles, population sizes and invasion histories? How might you expect these characteristics to have resulted in different ecological networks?

Line 288: should read "...overarching patterns"

Lines 291-294: How do we know that this is true – that there is no effect of the invasive birds on the abundance of native species in this system?

Methods

Lines 343-344: It is very interesting that there are several other species of introduced parrots in this system. I agree it makes sense to exclude them from the analysis, but do you expect these populations could increase in abundance? If so, would it make sense to speculate on their role as seed dispersers/predators considering your findings?

Tables

Table 1: There are many single observations in this table ($n=1$) for particular birds and plants. How does this affect your confidence in the ecological networks you have constructed?

Table 3: What is meant by "trophic facilitations"? Maybe reword for clarity?

Reviewer #2

(Remarks to the Author)

This was my first reading of the manuscript titled "Multi-layer networks reveal changes in plant-bird interactions driven by invasive species". I found the contents and the results of the manuscript highly interesting. The authors address a gap in the study of alien species, namely, their role in secondary invasions. Additionally, the paper also addresses interactions of species in detail. The study of interactions and through them functional traits is important while understanding the role of exotic species in novel ecosystems. Hence, I think the study will be interesting to a wide audience and contributes well to the literature. Additionally, I would also like to congratulate the authors on the amount of data that they have collected. Collecting field data over an entire year is not something that is often done. In doing so they have captured the variations in bird-plant interactions through the seasons. I also appreciate the detailed study of behaviour of the parakeets and their various interactions with the plants in terms of seed dispersal. That being said, below I highlight some major comments and then some minor comments regarding some of the decisions made in the MS.

Major comments:

1. Was only a single plant of each of the plant species observed? If so, it would be good to state this clearly and discuss what the possible shortcoming of observing a single plant of each species could be. If not, I would like to see how many individual plants of each species were sampled. And in the methods, it would be good to have an idea if you tried to balance this observational design somehow. I am basing this comment on my own experience from observing birds interact with fruiting plants in the field, certain plants seem to be favourites while the others of the same species even on the same street are visited when the favourite ones are depleted.
2. Line 137 – 140: How certain are the authors that the seeds that were handled by the invasive birds and then consumed by the native birds did in fact not have the potential to pass through their gut undigested anymore. How did the authors determine this? Was it through some other methods like looking at the faeces that are not mentioned in the paper? Additionally, how easy was it to observe epizoochory or stomatochory? Were the seeds always of such large sizes that they could be seen?
3. Line 169- 192: I am left wondering if species of exotic plants were more likely to be dispersed by one of the modes rather than the other. The heading talks about interaction of the parakeets with native and exotic plants, but the results focus more on the mode of dispersal.
4. Line 206 – 208: Are they also not benefitting native plants? Since you did not find a difference between the interactions of birds between native and exotic plants?
5. Line 212: Over here I do not like the use of the phrase "increased plant richness". As I understand the plants were already there but they did not show up in the interaction networks before the parakeets were included. I would change this to mean something accordingly.
6. Line 441: "assuming a normal error distribution", does this indicate a gaussian distribution? As I understand the response variable here was the number of species, which is a count and is not a continuous variable as such. In this case a Poisson distribution would be used rather than a Gaussian distribution.
7. Over all from the results that the number of exotic plants did not differ between the invasion and the pre-invasion networks,

and the fact that the parakeets seem to be urban species, I think the evidence for there being secondary invasions is weak. It would be interesting to see a more nuanced discussion of how the plant species are aiding frugivorous birds in an urban area where there is probably a large amount of plant loss taking place. Additionally, to believe the theory of secondary invasions it would be interesting to know how long the exotic plant species have been in the area, are the considered naturalised or invasive, how are their populations doing outside the area of introduction. Some data regarding these trends would help the discussion in my opinion. Or perhaps if the plant data are not available, some indications on data regarding the spread of the parakeets and discussion on this would help make the argument for secondary invasions stronger.

8. Lastly, I agree with the authors argument that about evolutionary origins helping the spread of exotic species through other exotics. With respect to this, it would be interesting to know how many of the exotic plants in the study area share evolutionary origins with the parakeet species and what impact this has on the networks. Perhaps this is a separate study questions altogether and I understand if the authors do not want to go into detail in this for the current MS. But another column in the supplementary table regarding the species origins might even be enough to give an idea.

Minor comments:

Line 15: "most invasive" to me this feels like a very strong claim. Later in the MS the authors call them wide spread invaders. Something like this sounds better. I would ask the authors to reconsider their phrasing here.

Line 19: "Acting a" seems like a typo

Line 59: "sometimes complicated and overly simplistic" I am not sure what this means. Perhaps complicated OR overly simplistic?

Line 59: for example... Why does antagonism not have a citation if mutualism does?

Line 76: extra bracket

Line 119: The sentence comes abruptly and I can try to infer the meaning but it is unclear. Consider rephrasing.

Line 120-123: Is this the summary for parakeets or all bird species?

Line 217: In some places parrot is used and in others parakeets. When not referring to a species in particular it would be good to stick to one or another.

Line 355: How many of the 35 plants were native or exotic? Would be nice to see here rather than having to dig through the supplements.

Table 1: Why are some of the results in bold text?

Version 1:

Reviewer comments:

Reviewer #1

(Remarks to the Author)

Thank you for the opportunity to review the revised version of this manuscript. I find the manuscript much improved and I commend the authors for their careful and thorough response to reviewer comments. With the addition of methodological detail, and nuanced discussion of limitations and opportunities for future research in the discussion, I find the science defensible and exciting. I only have one suggestion for improvement at this stage.

I might suggest modifying how germination success is currently addressed in the discussion. I think this is in fact critical to parakeets either posing a risk or a benefit to native plant communities. I understand that collecting data on this was beyond the scope of your study but the current text seems to diminish the importance of considering where seeds are dispersed to. There is no mutualistic relationship if parakeets are always depositing seeds in unfavorable microclimates for germination and growth.

Lines 348-352:

Current text: Although this aspect lies far beyond the scope of our study and is heavily dependent on local, often non-generalizable conditions, it does not invalidate the role of these species as plant mutualists. Rather, it underscores the added complexity involved in accurately assessing the ecological risks posed by invasive species.

My suggestion: "Although measuring germination success was beyond the scope of our study, understanding whether parakeet-mediated seed dispersal leads to plant germination and growth is ultimately critical to assessing the potential ecological risks or benefits posed by these invasive species."

Reviewer #2

(Remarks to the Author)

I have now read through the MS a second time after the authors included changes based on my first comments. I think all of my concerns were very well adjusted by the authors. I appreciate the effort the authors have put into making the changes, however, as a minor concern the discussion section is now rather lengthy. I will leave it up to the authors to improve it if they would like. From a scientific point of view I have no other concerns and would support the publication of the MS.

Manuscript COMMSBIO-25-1272-T

Dear Dr. Repetto,

Thank you very much for the positive response from the journal regarding our article, *COMMSBIO-25-1272-T*, entitled "**Multi-layer networks reveal changes in plant-bird interactions driven by invasive species**". We sincerely appreciate the constructive feedback provided by the reviewers, which has greatly improved the quality of our manuscript.

In the attached rebuttal letter, we have detailed all the revisions made to the text, addressing each comment point-by-point. To summarise the most relevant changes, we have:

1. **Substantially clarified the methods**, particularly regarding spatial scope, plant and site selection, sampling effort, and how we classified interactions as mutualistic or antagonistic. We also added new figures and supplementary tables to improve transparency and replicability.

2. **Revised the interpretation of parakeet-driven seed dispersal**, explicitly acknowledging that dispersal does not equate to successful recruitment, especially in urban settings, and discussing the ecological limitations and implications of this.

3. **Refined our treatment of secondary invasions**, framing them as a potential risk rather than a certainty, and discussing the ecological relevance of parakeet-facilitated dispersal of exotic plants. We also added data on species origins to support further insight into possible biogeographic affinities.

4. **Included new data on the geographic origins of bird and plant species** (Supplementary Table S3), allowing readers to assess potential biogeographic affinities that may help explain certain interaction patterns.

5. **Updated several statistical analyses**, changing the distribution of one model from Gaussian to Poisson (where appropriate for count data) and clarified assumptions and transformations used in others.

6. **Enhanced overall clarity and readability** of the manuscript by rewording ambiguous phrases, improving figure legends, and expanding explanations in key parts of the Results and Discussion.

In addition to these major changes and to addressing all other suggestions from the reviewers, we have performed a general review of the text, updating some sentences to improve clarity and readability. Finally, we have converted all graphs depicting a single point value (e.g., mean) with error bars to box-plots and prepared a Supplementary Data file (in excel format) with all source data for graphs.

We hope the current version is now suitable for publication in *Communications Biology* and would like to express our gratitude to you and the reviewers for your time and valuable insights. Please don't hesitate to contact us if any additional modifications are required.

Best regards,

Martina

(On behalf of all authors)

Reviewer #1: Thank you for the opportunity to review this interesting and well written paper on the ways that invasive parrots can disrupt and restructure plant-animal interactions. I think this paper will be of broad interest given that the focal species (two species of parakeets) have been introduced and established in dozens of countries across continents. More generally, understanding how invasive species change ecological processes such as seed predation and dispersal, including their potential to catalyze secondary invasions, is both ecologically interesting and critical to maintaining native biodiversity. Nonetheless, I do have the following concerns and suggestions for improvement.

RESPONSE: We sincerely thank the reviewer for his/her positive feedback on our manuscript. We are delighted that he/she found the study as compelling as we did.

Major Comments

Reviewer #1: My largest concern is that there are many aspects of the methodological approach that are unclear, limiting my ability to evaluate the findings, and the ability of others to replicate this study. For example, was the spatial scope of your study the entire city of Seville? How were the individual plants selected for observation? Were these plants/locations randomly selected, stratified across the city or in relation to the abundance of the invasive parakeets, etc.? What are your sample sizes – e.g., the number of individuals of each plant species observed? How did you select these sample sizes to ensure sufficient power to detect patterns? Finally, how did you track the fate of fruits/seeds and determine whether interactions were mutualistic or antagonistic? This seems extremely challenging given these birds are highly mobile and fast flying.

RESPONSE: We understand that we may not have been sufficiently clear in describing our field methods, and we acknowledge that this may have limited the ability to fully evaluate the study and its replicability. We also recognize that these same questions could arise among readers, so we have revised the manuscript to expand and clarify all these aspects:

- The spatial scope of the study encompassed the entire urban area of the city of Seville and its immediate surroundings. This is now clearly illustrated in a newly added figure (Supplementary Fig. 2), which shows the locations of all individual plants sampled across the city.

To ensure spatial representativeness while considering site accessibility, we stratified the study area into distinct sectors, including all major parks, gardens, and adjacent rural zones with more natural vegetation. Based on previous fieldwork, we knew that not all individual plants would be visited by birds. Therefore, within each sector, we surveyed available areas by walking to detect plant individuals being visited by birds. Once a plant with foraging activity was detected, we began detailed behavioural observations. Our approach prioritized maximizing interaction detection, so we monitored as many individuals of each plant species as possible. For plant species that attracted very few birds, we conducted additional focused observations to ensure adequate sampling and confirm that low interaction rates were not an artifact of sampling effort. In some of these cases, even if interaction accumulation curves did not stabilize, we retained the data when the results aligned with ecological expectations (e.g., fruit types not commonly consumed by native birds). Sample sizes are now explicitly detailed for each plant species in Supplementary Table 1. These sample sizes were chosen based on a balance between taxonomic and spatial coverage and logistical feasibility, guided by interaction accumulation curves (Supplementary Fig. 3) to ensure sufficient representation.

The tracking of fruit and seed fate was based on systematic focal observations. During these sessions, we closely recorded the behavior of birds interacting with fruits, noting whether fruits and/or seeds were predated, dropped, swallowed, or carried away and how. A detailed description of interaction types was provided in the Supplementary Material, which we believe may have led to some confusion for the reviewer. To address this and improve clarity for future readers, we have expanded this section of the Methods in the revised manuscript to include a more detailed explanation of the criteria used to classify each interaction. It is important to emphasize that accurately determining the fate of all fruits and seeds handled by birds is inherently challenging. This is why the sample size of individual plants may appear small in some cases —we prioritized extensive observation time to each plant, often with two simultaneous observers, to ensure the reliability of our data.

We are grateful to the reviewer for these constructive suggestions, which have helped us significantly improve the clarity, transparency, and replicability of our methodological description.

Figure 2. Map of the study area (outlined in red) covering the metropolitan area of Seville and surroundings (southern Spain) (indicated by a red dot in the inset map). The locations of monitored plants (i.e., where plant-bird interactions and dispersal mechanisms were recorded) are shown as blue dots.

I would be careful to clarify that dispersal does not necessarily result in successful germination, growth and survival of seeds/seedlings. Particularly in urban systems, random dispersal vs dispersal into particular microclimates may be critical for plant recruitment. As such, I would be cautious in stating that parakeets could augment or replace functions performed by native species at lower densities.

RESPONSE: We understand the reviewer's point. However, it is important to mention that this issue is equally applicable to native birds in urban environments, not only to parakeets. While it is true that equating dispersal with subsequent seedling recruitment is incorrect, studies on mutualistic networks generally focus on the first step: the movement of propagules away from the mother plant, which is the essential prerequisite. Nevertheless, addressing the reviewer's comment, we have incorporated a few sentences in the Discussion reasoning this situation, which further complicates

predicting the impact of invasive species and ecological systems in human dominated areas such as urban landscapes (lines 342-351): “To truly understand the extent of these impacts, it is also important to evaluate the effectiveness of seed dispersal in the urban environments these species typically inhabit. While previous research confirms that parrots are legitimate seed dispersers^{39-42,66-67,69}, dispersal in human-dominated environments or involving exotic plant species, which may sometimes be sterile, does not necessarily lead to successful germination, growth and survival of seedlings. Although this aspect lies far beyond the scope of our study and is heavily dependent on local, often non-generalizable conditions, it does not invalidate the role of these species as plant mutualists. Rather, it underscores the added complexity involved in accurately assessing the ecological risks posed by invasive species.”

Minor Comments

Abstract

Lines 17-18: Unless this is discouraged by the journal, I suggest adding a methods sentence that indicates where and how this question was addressed.

RESPONSE: We have added a sentence to briefly explain how we obtained the data to answer our research questions (lines 17-19): “To do this, we collected field data over a full annual cycle in an area inhabited by both species, accumulating 288 hours of observations and tracking the fate of 24,561 fruits from 576 individual plants.”

Line 19: Should read “Acting as both...”

RESPONSE: Corrected.

Introduction

Line 62: Is this mutualism if the bird does not receive a benefit? Maybe clarify that this is mutualism only if the bird consumes some of the fruit but does not damage the seed?

RESPONSE: We have explicitly stated that these interactions qualify as mutualisms only when birds derive a net reward from them (lines 62-65): “For example, seed eaters may act as predators when destroying seeds (antagonism), but can also as dispersers by plucking whole fruits from the mother plant and dropping intact seeds at a distance after consuming the pulp or even some seeds (mutualism²⁰).”

Lines 78-82: I think it would be helpful to include a global map that shows the native and introduced ranges of your focal species, with a symbol to indicate your study site. This could go in supplemental materials if there is not room in the main text. This could help illustrate the potential widespread relevance of your study.

RESPONSE: As suggested by the reviewer, we have included in the Supplementary Material (Supplementary Fig. 1) a global map with the distribution (native and invasive) of ring-necked and the monk parakeets.

Lines 94-96: Is it correct that you only collected 1 year of data? I am confused as here you state data was collected over a full annual cycle but later (Line 200) you refer annual monitoring, which suggests monitoring over multiple years. If the study was limited to one year that is understandable given the scope of the effort within this year, but I think it is important to discuss the potential limitations of this approach. Is there reason to believe that inter-annual variability in climate or other factors could impact bird and plant populations, phenology of foraging and fruiting, and species interactions? Also, were you able to look at the effect of seasons on species interactions within your study year?

RESPONSE: We appreciate the reviewer's thoughtful comment regarding the temporal scope of our study. Indeed, our work was based on a single year of field data, and we fully acknowledge in the updated Discussion that this introduces limitations, particularly concerning interannual variability in climate, resource availability or species abundances (lines 333-342): "Moreover, we also recognize that our single-year study design cannot capture inter-annual variability in climate, resource availability or species abundances, all factors that may influence plant-bird interactions and, by extension, network dynamics. Although our dataset provides exceptional temporal resolution, capturing a complete phenological cycle of plants and birds, it may still overlook important sources of ecological variation, including fruiting phenology shifts, extreme climatic events (e.g., droughts, frosts), changes in bird foraging behaviours or rare phenomena like masting or irruptive migrations³⁹. Future multi-year studies will be crucial to disentangle transient environmental effects from the persistent impacts of invasive parakeets on ecological networks, especially in the context of accelerating global change."

Discussion

Lines 232-233: Can you speak further to any differences between the two parakeet species in regard to the interaction networks? This is referenced again on lines 316-318, but not discussed in any detail. In what ways do these two species have different ecological roles, population sizes and invasion histories? How might you expect these characteristics to have resulted in different ecological networks?

RESPONSE: We have included a brief discussion following the reviewer comment. However, to not extend the manuscript unnecessarily, we refer the reader to the study that discussed potential differences in the invasion histories of both species that result in their population unbalance (lines 248-256): “While we lack comprehensive data on the relative abundance of all bird species in the study community to explicitly assess the relevance of neutral processes in driving nestedness⁶⁰, the large populations of both invasive parakeets may help explain their strong influence on network structure. Since their introduction in Seville in the 1990s, their numbers have steadily increased⁶¹, in line with broader European trends⁶²⁻⁶³. However, their population sizes differ locally -likely due to distinct invasion histories^{61,64} and, together with their broad but different diets and foraging strategies (with rose-ringed parakeets being more arboreal) may account for their differing roles in shaping network architecture and functionality.”

Line 288: should read “...overarching patterns”

RESPONSE: Done.

Lines 291-294: How do we know that this is true – that there is no effect of the invasive birds on the abundance of native species in this system?

RESPONSE: We have revised this section of the discussion to improve clarity and have softened the tone of the original sentence. In this revised section we better clarify the rationale and limitations of our approach. In particular, previous evidence suggests that effects of invasive bird species in native species abundance tends to ameliorate with time and, in the particular case of our study area, we have no evidence of declines of native bird species following parakeet introductions. Here is the revised paragraph (lines 316-333): “Understanding how invasive species alter ecological networks is inherently complex, as it requires comparisons with scenarios in which those species are absent.

Ideally, this would involve analysing independent ecological networks along invasion gradients or comparing interactions before and after the establishment of the invaders. However, such situations are rare and often confounded by other ecological factors, making this approach logistically unfeasible. To overcome these problems, researchers often rely on alternative methods, such as network meta-analyses to identify overarching patterns¹⁸, species-level interaction analysis⁸³, or simulations using null models to construct hypothetical baseline scenarios⁶⁷. In this study, we simulate a pre-invasion network by excluding interactions involving invasive species and their associated facilitative processes, using data collected at a single site and time point to minimize variability in species abundances. Although previous studies have shown that parakeets can reduce feeding rates among native birds at feeding stations⁸⁴⁻⁸⁵, this interference tends to decline over time as natives habituate to their presence⁸⁴. In our study area, where parakeets have established since the 1990s, there is no direct or indirect evidence that they have displaced or reduced the abundances of the native species included in our network. Therefore, our approach would offer a practical starting point for investigating the effects of invasive species, while future studies collect data from nearby uninvaded areas to more directly evaluate their impact on network functioning.”

Methods

Lines 343-344: It is very interesting that there are several other species of introduced parrots in this system. I agree it makes sense to exclude them from the analysis, but do you expect these populations could increase in abundance? If so, would it make sense to speculate on their role as seed dispersers/predators considering your findings?

RESPONSE: We have briefly discussed this point as follows (lines 269-275):

“Although not included in our analyses due to their low numbers, the blue-crowned parakeet (*Thectocercus acuticaudatus*) is currently present in the study area. Given that all parrots share similar feeding behaviours, it is likely that this species contributes similarly to the dynamics described for the two focal species. However, as we observed differences between the rose-ringed and monk parakeets, it is also possible that such differences exist with the blue-crowned parakeet, and therefore, its role should be investigated in future studies.”

Tables

Table 1: There are many single observations in this table ($n = 1$) for particular birds and plants. How does this affect your confidence in the ecological networks you have constructed?

RESPONSE: As the reviewer correctly points out, many interactions in the table are represented by a single seed ($n = 1$), highlighting their limited quantitative importance within the network. In qualitative networks, such single observations would have disproportionately influenced metrics and the assessment of species' roles, since all interactions are weighted equally regardless of frequency. However, in our quantitative approach, these low-frequency interactions are appropriately weighted, minimizing their impact on network structure and species-level metrics. While these interactions contribute little in numerical terms, acknowledging them is still important—they represent rare but real links within the community and help capture the full spectrum of ecological interactions occurring in the system.

Table 3: What is meant by “trophic facilitations”? Maybe reword for clarity?

RESPONSE: We have clearly explained facilitation in the table caption for all readers not familiarized with this term (lines 979-983: “**Table 4.** Food facilitation events recorded during the study period. Facilitation is defined as any interaction in which a species -primarily parakeets, Iberian magpies *Cyanopica cooki* or humans- enhances fruit or seed access for other bird species by discarding, partially consuming or modifying food items, thereby making them more available, accessible, or easier to consume”).

Reviewer #2:

This was my first reading of the manuscript titled “Multi-layer networks reveal changes in plant-bird interactions driven by invasive species”. I found the contents and the results of the manuscript highly interesting. The authors address a gap in the study of alien species, namely, their role in secondary invasions. Additionally, the paper also addresses interactions of species in detail. The study of interactions and through them functional traits is important while understanding the role of exotic species in novel

ecosystems. Hence, I think the study will be interesting to a wide audience and contributes well to the literature. Additionally, I would also like to congratulate the authors on the amount of data that they have collected. Collecting field data over an entire year is not something that is often done. In doing so they have captured the variations in bird-plant interactions through the seasons. I also appreciate the detailed study of behaviour of the parakeets and their various interactions with the plants in terms of seed dispersal. That being said, below I highlight some major comments and then some minor comments regarding some of the decisions made in the MS.

RESPONSE: We would like to sincerely thank the reviewer for his/her kind and encouraging words, highlighting the significance of our work and the importance of the exhaustive data collection involved. Such recognition is truly motivating and reinforces the value of the detailed efforts we have put into this study.

Major comments:

1. Was only a single plant of each of the plant species observed? If so, it would be good to state this clearly and discuss what the possible shortcoming of observing a single plant of each species could be. If not, I would like to see how many individual plants of each species were sampled. And in the methods, it would be good to have an idea if you tried to balance this observational design somehow. I am basing this comment on my own experience from observing birds interact with fruiting plants in the field, certain plants seem to be favourites while the others of the same species even on the same street are visited when the favourite ones are depleted.

RESPONSE: Contrary to what might have been inferred, we did not monitor only one individual per plant species. Instead, we observed a variable number of individual plants per species, depending on their local availability and attractiveness to birds. Since this is a recurring concern (Reviewer 1 similarly highlighted the importance of specifying sample sizes), we have now made this information explicit (Supplementary Table 1). These sample sizes were determined to ensure broad taxonomic and spatial coverage, while remaining feasible within the logistical constraints of fieldwork. The rationale for determining sample sizes was guided by species accumulation curves (Supplementary Fig. 3), which we used to assess sampling completeness and ensure sufficient representation of plant–bird interactions. Besides, we have also included a Figure

(Supplementary Fig. 2) so readers can clearly see the spatial scope of the study, showing the locations of all individual plants sampled across the city.

To capture the spatial distribution of bird species while considering site accessibility, individual plants were selected by stratifying the study area into distinct sectors, including all major parks, gardens, and adjacent rural areas with more natural vegetation. Based on previous fieldwork, we knew that not all individual plants would be visited by birds (as also pointed by the reviewer). Therefore, we surveyed each area by walking through it and identifying target plants; once bird foraging activity was detected, we began recording interactions. We monitored as many individuals of each plant species as possible to maximize detection of interactions. In some cases, particularly for plant species that attracted very few birds, we conducted additional observations to ensure that the lack of stabilization in interaction accumulation curves was not due to insufficient sampling. For these species, we even retained the data when the patterns aligned with ecological expectations (e.g., fruits not typically consumed by native birds). To clearly show the distribution of our observations, we have included a new figure in the Supplementary Material (Fig. 2) with the study area and the location of the different individual plants.

2. Line 137 – 140: How certain are the authors that the seeds that were handled by the invasive birds and then consumed by the native birds did in fact not have the potential to pass through their gut undigested anymore. How did the authors determine this? Was it through some other methods like looking at the faeces that are not mentioned in the paper? Additionally, how easy was it to observe epizoochory or stomatochory? Were the seeds always of such large sizes that they could be seen?

RESPONSE: While we did not directly assess seed viability post-consumption by native birds in each case, both our field observations and the cited literature indicate that the ingestion of unripe fruits generally results in seed destruction rather than successful endozoochorous dispersal. For this reason, we assumed the same outcome applies when such unripe seeds are consumed after being made accessible through parakeet-mediated facilitation. In the revised version of the manuscript, we now explicitly state this assumption in the main text, as it was previously only mentioned in the Supplementary and may have been overlooked (lines 440-446): “Seeds that were swallowed whole were generally considered dispersed by endozoochory unless damaged while passing

through the bird's digestive tract. Such instances, confirmed through faecal and regurgitate analysis, included: 1) ingestion of unripe fruits with unprotected seeds⁶⁶, 2) ingestion of ripe fruits with small seeds (< 5 mm) and soft coats (seed hardness = 1; Supplementary Table 1) by pigeons and doves¹⁰³⁻¹⁰⁴ and 3) ingestion of acorns by the wood pigeon (*Columba palumbus*), whose gizzard completely crushed them¹⁰⁵.”

Regarding epizoochory and stomatochory, both dispersal methods were relatively easy to observe during our fieldwork, particularly stomatochory. Observations were carried out from strategically located vantage points that were not far from the focal plants, and we used appropriate optical equipment, including binoculars and high-powered spotting scopes, to ensure detailed visual monitoring. Because we were aware of the possibility of these dispersal mechanisms (see previous studies from our group in this regard, cited in the main manuscript with references: 39-42,66-67,6), we paid special attention to identifying them, especially in plant species known to be more frequently involved in such interactions. In the case of small seeds dispersed via epizoochory—particularly in berry-like fruits—we employed powerful scopes to confirm the attachment of seeds to the birds' beaks or faces, which can be difficult to detect with the naked eye. Thus, our targeted approach and experience with the system enabled reliable detection of these less common, but ecologically relevant, dispersal events.

3. Line 169- 192: I am left wondering if species of exotic plants were more likely to be dispersed by one of the modes rather than the other. The heading talks about interaction of the parakeets with native and exotic plants, but the results focus more on the mode of dispersal.

RESPONSE: While this part of the Results indeed describe dispersal modes, it also addresses broader patterns of interactions between parakeets and both native and exotic plant species, and how these interactions influence the overall structure and dynamics of the networks.

In response to the reviewer's specific question, we have now included a brief analysis comparing the primary dispersal modes for native and exotic plants. Although both plant groups are predominantly dispersed via endozoochory, this mode is significantly more frequent among exotic species. This is now stated explicitly in the results section (lines 197-200): “However, since both native and invasive bird species interact with exotic plant species, the most common dispersal mechanism for these plants remains

endozoochory overall (81% and 76% of fruits dispersed via endozoochory in exotic and native plants, respectively; $\chi^2 = 20.48$, $DF = 1$, $P < 0.0001$).”

4. Line 206 – 208: Are they also not benefitting native plants? Since you did not find a difference between the interactions of birds between native and exotic plants?

RESPONSE: We have reworded this sentence to make it clear that both plants and birds can be benefitted (lines 214-219): “Although these alterations may benefit certain native plants and birds by increasing interaction opportunities, they also introduce instability and unpredictability into network dynamics. In particular, by enabling the dispersal of exotic plants previously excluded from local networks, parakeets may inadvertently facilitate secondary invasions, amplifying long-term ecological risks and challenging conservation efforts in increasingly human-dominated landscapes.”

5. Line 212: Over here I do not like the use of the phrase “increased plant richness”. As I understand the plants were already there but they did not show up in the interaction networks before the parakeets were included. I would change this to mean something accordingly.

RESPONSE: We have reworded this sentence accordingly (lines 222-224): “The entrance of invasive parakeets significantly increased the number of plant species and interactions in the network, especially within the antagonistic subnetwork, by incorporating species not previously involved in interactions.”

6. Line 441: “assuming a normal error distribution”, does this indicate a gaussian distribution? As I understand the response variable here was the number of species, which is a count and is not a continuous variable as such. In this case a Poisson distribution would be used rather than a Gaussian distribution.

RESPONSE: We thank the reviewer for this comment. Indeed, a Poisson GLM is the correct model in this context. We have implemented this model in the updated version, and updated the results accordingly (lines 521-526): “To understand how parakeets interacted with plants based on their origin, we used Generalized Linear Models to compare the number of bird species (Poisson error distribution, log link function) interacting with each plant and the weight (log-transformed; normal error distribution,

identity link function) of these interactions –covering both antagonisms and mutualisms and including the specific dispersal mechanisms involved- between native and exotic plants in the pre-invasion and invasion network (fixed factors)”. Note that the main outcomes are maintained.

7. Over all from the results that the number of exotic plants did not differ between the invasion and the pre-invasion networks, and the fact that the parakeets seem to be urban species, I think the evidence for there being secondary invasions is weak. It would be interesting to see a more nuanced discussion of how the plant species are aiding frugivorous birds in an urban area where there is probably a large amount of plant loss taking place. Additionally, to believe the theory of secondary invasions it would be interesting to know how long the exotic plant species have been in the area, are they considered naturalised or invasive, how are their populations doing outside the area of introduction. Some data regarding these trends would help the discussion in my opinion. Or perhaps if the plant data are not available, some indications on data regarding the spread of the parakeets and discussion on this would help make the argument for secondary invasions stronger.

RESPONSE: We thank the reviewer for this thoughtful and constructive comment, and we agree that our discussion benefits from a more nuanced treatment of the potential for secondary invasions. While it is true that both parakeet species are primarily urban in origin, their populations are currently expanding, and in some regions, they have begun to colonize peri-urban and even rural environments (Hernández-Brito, D., Blanco, G., Tella, J.L. & Carrete, M. 2020. A protective nesting association with native species counteracts biotic resistance for the spread of an invasive parakeet from urban into rural habitats. *Journal of Zoology* 17, 13; Hernández-Brito, D., Carrete, M. and Tella, J.L. 2022. Annual Censuses and Citizen Science Data Show Rapid Population Increases and Range Expansion of Invasive Rose-Ringed and Monk Parakeets in Seville, Spain. *Animals*). This spatial expansion broadens their ecological footprint and enhances their potential to disperse seeds beyond city boundaries, thereby increasing the relevance of their frugivory in a broader ecological context. Indeed, during our study, both species were observed foraging in areas beyond urban centers, particularly in rural gardens and dispersed dwellings where many exotic plants are commonly cultivated. Given their high mobility, parakeets may serve as effective vectors of both native and exotic plant

species from these human-dominated habitats into natural or semi-natural areas. This capacity opens the door to secondary invasions, especially for exotic plants that are not typically dispersed by native frugivores. While we do not provide detailed historical or demographic data on the introduction and spread of the exotic plant species in our study area, many are widely cultivated, several are naturalized, and a few are recognized as invasive elsewhere in the Mediterranean basin. We acknowledge that a thorough assessment of these plant species' invasive potential lies beyond the scope of our current study. Our objective here is to demonstrate that parakeet-mediated dispersal facilitates novel interactions that could, under certain circumstances, contribute to secondary invasions. As the reviewer rightly notes, secondary invasions are complex phenomena shaped by local species pools, dispersal mechanisms, and landscape structure. Predicting them with certainty is inherently difficult. However, it is important to emphasize that parakeets are introducing a new dispersal dynamic—one that enables the movement of plant species, particularly exotics, which were not previously dispersed by native birds. This alone is ecologically significant and merits attention from a risk assessment perspective. In the revised manuscript, we have clarified these points and reframed the secondary invasion scenario as a possibility rather than a definitive outcome. Nonetheless, we stress the importance of monitoring this process, especially given the ongoing expansion of both invasive parakeets and the exotic flora they interact with in urban and surrounding landscapes.

8. Lastly, I agree with the authors argument that about evolutionary origins helping the spread of exotic species through other exotics. With respect to this, it would be interesting to know how many of the exotic plants in the study area share evolutionary origins with the parakeet species and what impact this has on the networks. Perhaps this is a separate study questions altogether and I understand if the authors do not want to go into detail in this for the current MS. But another column in the supplementary table regarding the species origins might even be enough to give an idea.

RESPONSE: Thank you for your thoughtful suggestion. We agree that exploring the evolutionary or biogeographic affinities between exotic plants and parakeets is a very interesting direction for future research. While a detailed analysis of species origins is beyond the scope of the current manuscript, we have now included a new table (Supplementary Table S3) that provides the geographic ranges of the bird species

included in the study. We hope this addition offers a useful first step towards understanding these potential biogeographic links.

Minor comments:

Line 15: “most invasive” to me this feels like a very strong claim. Later in the MS the authors call them wide spread invaders. Something like this sounds better. I would ask the authors to reconsider their phrasing here.

RESPONSE: We appreciate the reviewer’s suggestion and agree that “most invasive” may overstate the status of these species in a global context. To better reflect their documented distribution and ecological impact, we have revised the sentence as follow (lines 15-17): “Here, we investigate how the *two most widespread invasive parrots* –the rose-ringed parakeet *Psittacula krameri* and monk parakeet *Myiopsitta monachus*– affect plant-bird interaction networks using a multilayer framework.”

Line 19: “Acting a” seems like a typo

RESPONSE: Corrected.

Line 59: “sometimes complicated and overly simplistic” I am not sure what this means. Perhaps complicated OR overly simplistic?

RESPONSE: Thank you for pointing out the ambiguity in our wording. We have revised the sentence to clarify that we mean the classification of species into strictly mutualistic or antagonistic roles is both difficult due to ecological complexity and potentially reductive, as it may not fully capture the dynamic nature of species interactions. The new phrasing aims to express this more clearly (60—62): “However, classifying species as purely mutualistic or antagonistic can be both conceptually challenging and overly simplistic, as it may overlook the complexity of plant-animal interactions.”

Line 59: for example... Why does antagonism not have a citation if mutualism does?

RESPONSE: Sorry for this error. Our intention was for the citation to support the full example, encompassing both the antagonistic and mutualistic outcomes of seed-eater behavior. However, to avoid ambiguity and ensure clarity, we have revised the sentence

slightly and repositioned the citation to clearly indicate it refers to the entire example (lines 62-65): “For example, seed eaters may act as predators when destroying seeds (antagonism), but can also as dispersers by plucking whole fruits from the mother plant and dropping intact seeds at a distance after consuming the pulp or even some seeds (mutualism)²⁰.”

Line 76: extra bracket

RESPONSE: Deleted.

Line 119: The sentence comes abruptly and I can try to infer the meaning but it is unclear. Consider rephrasing.

RESPONSE: We agree with the reviewer that the original sentence appeared abruptly and could be confusing. To improve clarity and readability, we have now rephrased it in the revised manuscript (lines 123-125):” Note that, for simplicity, we treated three plant genera, each comprising two morphologically similar species, as single species in our analyses (see Methods).”

Line 120-123: Is this the summary for parakeets or all bird species?

RESPONSE: We thank the reviewer for pointing this out. The summary provided refers to the outcomes for all bird species observed during the study period, not just the parakeets. To avoid confusion, we have now clarified this in the manuscript text (lines 125-132): “Among the fruits monitored across all bird species, 19% were defleshed, 5% were wasted and remained in the ground without being consumed during the focal sampling, 44% were predated and 32% were dispersed by endozoochory (78%), epizoochory (2%) or stomatochory (20%). This information was used to build an antagonistic-mutualistic multilayer network (hereafter, “invasion network”). By removing the parakeets and all associated interactions and species that only interacted with them or were facilitated by them, we simulated the most likely scenario prior to the arrival of the invaders (hereafter, “pre-invasion network”) for comparison.”

Line 217: In some places parrot is used and in others parakeets. When not referring to a species in particular it would be good to stick to one or another.

RESPONSE: We appreciate the reviewer’s attention to consistency in terminology. In our manuscript, we use “*parrot*” when referring broadly to members of the order *Psittaciformes*, and “*parakeet*” specifically when discussing our focal invasive species—*Psittacula krameri* and *Myiopsitta monachus*. To ensure clarity, we have double-checked the text to make sure this distinction is consistently maintained throughout.

Line 355: How many of the 35 plants were native or exotic? Would be nice to see here rather than having to dig through the supplements.

RESPONSE: Thank you for pointing this out. We realized there was a mistake in the text: we originally stated that 35 plant species were included, but one species was later removed from the analyses due to having too few interactions, so the correct number is 34. Note that this was only a typo in the writing – the analyses were performed with the correct number of species. We have now corrected this in the manuscript and clearly stated the number of native and exotic plant species (lines 119-125): “During the study period, we accumulated ca. 288 h of observations, tracking the fate of 24,561 fruits from 576 individual plants (Supplementary Fig. 2). This effort resulted in the documentation of 243 unique interactions involving 24 bird and 13 native and 21 exotic plant species (Supplementary Tables 1-3) with sampling effort sufficiently exhaustive to ensure that the majority of interacting species were reliably registered (Supplementary Fig. 3).”

Table 1: Why are some of the results in bold text?

RESPONSE: The values in bold indicate observed network metrics that significantly differ from those expected under the null models. We have now clarified this in the caption of Table 1 to avoid confusion (lines 818-821): “**Table 1.** Network parameters obtained for the multilayer networks and their two constituent layers -the antagonistic and mutualistic subnetworks- before (pre-invasion) and after (invasion) the establishment of invasive parakeets in the system. Observed values that significantly differ from those expected under the null models are shown in bold.”